# Versatile strategy for homogeneous drying patterns of dispersed particles

Marcel Rey[1,2,3], Johannes Walter[1,2], Johannes Harrer[1,2], Carmen Morcillo Perez [3], Salvatore Chiera[1,2], Sharanya Nair[1,2], Maret Ickler[1,2], Alesa Fuchs[1,2], Mark Michaud[1,2], Maximilian J. Uttinger[1,2], Andrew B. Schofield[3], Job H. J. Thijssen [3], Monica Distaso[1,2], Wolfgang Peukert [1,2] & Nicolas Vogel [1,2✉]

After spilling coffee, a tell-tale stain is left by the drying droplet. This universal phenomenon, known as the coffee ring effect, is observed independent of the dispersed material. However, for many technological processes such as coating techniques and ink-jet printing a uniform particle deposition is required and the coffee ring effect is a major drawback. Here, we present a simple and versatile strategy to achieve homogeneous drying patterns using surface-modified particle dispersions. High-molecular weight surface-active polymers that physisorb onto the particle surfaces provide enhanced steric stabilization and prevent accumulation and pinning at the droplet edge. In addition, in the absence of free polymer in the dispersion, the surface modification strongly enhances the particle adsorption to the air/ liquid interface, where they experience a thermal Marangoni backflow towards the apex of the drop, leading to uniform particle deposition after drying. The method is independent of particle shape and applicable to a variety of commercial pigment particles and different dispersion media, demonstrating the practicality of this work for everyday processes.

[1] Institute of Particle Technology (LFG), Friedrich-Alexander-Universität Erlangen-Nürnberg (FAU), Cauerstrasse 4, 91058 Erlangen, Germany. [2] Interdisciplinary Center for Functional Particle Systems (FPS), Friedrich-Alexander-Universität Erlangen-Nürnberg (FAU), Haberstrasse 9a, 91058 Erlangen, Germany. [3] School of Physics & Astronomy, The University of Edinburgh, Peter Guthrie Tait Road, Edinburgh EH9 3FD, UK. ✉email: nicolas.vogel@fau.de

The coffee ring effect is a complex physical process, where hydrodynamic flow, capillary effects and multiple interactions between the interfaces of a liquid, a substrate and the dispersed particles come into play. The coffee ring forms because the three-phase contact line is pinned to the substrate and preferential evaporation of the liquid occurs at this edge. This liquid is replenished from the droplet center, inducing a capillary flow towards the drying edge. Thus, any dispersed material is carried towards the droplet edge, forming the characteristic ring pattern[1–3]. While coffee ring patterns can be exploited for micro- and nanopatterning[4,5] or analytics[6,7], uniform drying patterns are generally preferred in technological applications, such as coating processes[8,9] or inkjet printing[10–13].

The importance of achieving uniform drying patterns has led to different strategies to suppress the coffee ring effect. Capillary flow can be reduced by an increase in viscosity due to the addition of soluble polymers[14–17], by gelation of the liquid[18] or by making the substrate porous[19]. Alternatively, Marangoni flows[20–23] can be invoked to at least partially counteract the capillary flow. Increasing particle–particle and particle–surface interactions[24] can cause aggregation and therefore increased sedimentation of the dispersed material[25] or electrostatic adsorption to the substrate[26]. Interactions of the dispersed material with tailored surfactants or ligands also lead to a more uniform drying behavior[27–30]. Finally, the shape of the particles itself has been shown to affect the coffee ring formation, with ellipsoidal particles causing a uniform drying behavior[31]. These procedures all require carefully tailored model systems, which limits the applicability in product manufacture.

Here we describe a simple and versatile strategy to overcome the coffee ring effect by modifying the surface of the dispersed particles with surface-active polymers.

## Results

**Polymers adsorbed to particles affect the drying behavior.** To demonstrate the effect, we start with a model system consisting of a polystyrene (PS) particle dispersion with a polyvinyl alcohol (PVA) polymer solution (Fig. 1a, b). The pure polystyrene reference dispersion exhibits a pronounced coffee ring after drying (Fig. 1c, Supplementary Movie 1). The addition of PVA increases the viscosity and thus leads to an improvement in the uniformity of the drying behavior compared to the pure dispersion, as known from literature[15] (Fig. 1d). However, confocal microscopy images taken during the drying process (Fig. 1, bottom) reveal that, similar to the pure dispersion used as reference, the majority of the particles still accumulate at the droplet edge. As a result, the drying pattern is not homogeneous and shows ill-defined ring-like features, although less pronounced compared to the unmodified sample. Surprisingly, a fully uniform drying pattern results when free polymer is removed from the dispersion by repeated centrifugation-redispersion cycles (Fig. 1e, Supplementary Fig. 1). In this case, only polymer chains that are physisorbed at the particle surfaces are present in the system. The absence of free polymer in the dispersion is evidenced by its surface tension, which approaches that of a pure PS dispersion (Fig. 1b, Supplementary Fig. 2). In contrast, the PVA-modified dispersion without washing steps shows a significant reduction in

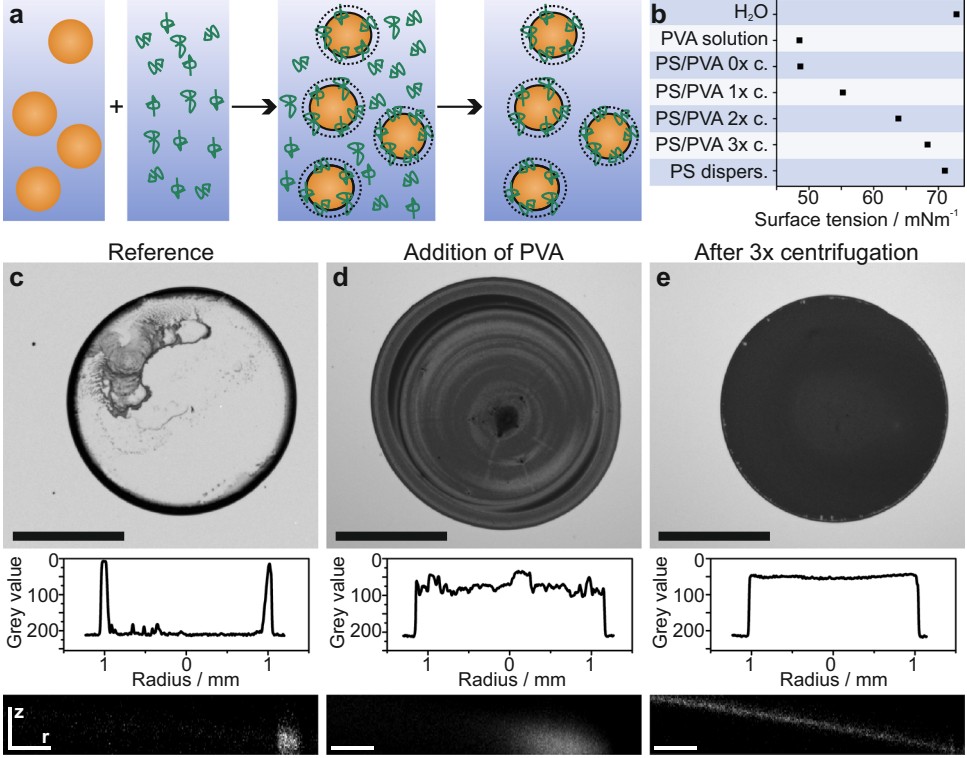

**Fig. 1 Mitigating the coffee ring effect by surface modification of dispersed particles. a** Schematic illustration of the process: An aqueous polystyrene (PS) particle dispersion is mixed with excess polyvinyl alcohol polymer (PVA) solution. The polymer partially adsorbs onto the particles. Centrifugation and redispersion removes excess, non-adsorbed polymer. **b** Surface tension of the different aqueous dispersions used to investigate the drying behavior. **c–e** Drying behavior of a reference colloidal dispersion (**c**), the same dispersion with addition of PVA (weight ratio 1:1) (**d**), and after removal of non-adsorbed PVA (**e**). In (**c–e**) top image: Optical micrograph of the dried dispersions. Scale bar: 1 mm. Middle image: Cross-sectional grey value distribution after drying. Bottom image: Confocal microscopy side projections (z–r plane) of the drying dispersions using fluorescently labelled PS particles. Scale bar: 25 μm.

surface tension to 47 mNm$^{-1}$, indicating that free polymer covers the air/water interface. As shown in the confocal microscopy images of Fig. 1e, a consequence of the removal of this free polymer is that the dispersed particles adsorb at the air/water interface of the drying droplet much more efficiently and a homogeneous drying pattern results.

Our results show similarities to the drying behavior of ellipsoids by Yunker et al.[31], which were prepared by mechanical stretching of PS particles embedded in a PVA foil[32] and subsequent dissolution and purification by centrifugation and redispersion in water[31]. Following that protocol, we could reproduce the uniform drying pattern of elliptical particles (Fig. 2a, Supplementary Movie 2). In contrast to these previous findings, however, a sample with spherical particles prepared by the same protocol, without the stretching step, shows a similar, uniform drying pattern (Fig. 2b, Supplementary Fig. 3, Supplementary Movie 3). When the particle dispersions are subsequently cleaned by centrifugation and redispersion in isopropyl alcohol (IPA)/water mixtures, which efficiently removes PVA[32], both elliptical and spherical particle dispersions show a pronounced coffee ring after drying (Fig. 2d, e, Supplementary Fig. 3, Supplementary Movie 4, 5, Supplementary Discussion 1). The observations indicate that the presence of PVA chains physisorbed onto the particle surface, rather than the particle shape itself, dominates the drying behavior.

We quantify the presence of PVA adsorbed to the PS particles using dynamic light scattering. We measure an increase in hydrodynamic diameter $D_H$ of 51 nm from $(328 \pm 7)$ nm for pure particles to $(379 \pm 4)$ nm for PVA-water washed particles, highlighting the presence of a PVA shell. After washing in IPA/water mixtures, a $D_H = (328 \pm 6)$ nm indicates the removal of the physisorbed PVA shell (Supplementary Fig. 4). We further characterize the composition of the PVA shell from the sedimentation properties of the particles in $H_2O/D_2O$ mixtures using analytical centrifugation (details in Methods, Supplementary Figs. 5–7). We determine a PVA shell thickness of 30 nm and a PVA volume fraction of 4.9%, corresponding to a PVA surface coverage of 2.3 mgm$^{-2}$, in agreement with previous work[33]. The interpretation of the presence and absence of PVA agrees with cryo-scanning electron microscopy (SEM) observations from literature[34]. To further underline the impact of surface-active polymers on a particle surface and demonstrate that the effect is not related to any particular property of PVA, or the particle fabrication process, we compare the drying behavior of silica particles with grafted linear poly(N-isopropyl acrylamide) (Fig. 2c) to the one of pure silica particles (Fig. 2f). While pure silica particle dispersions form the expected coffee ring pattern (Fig. 2f), particles with grafted polymer chains display a uniform drying pattern (Fig. 2c), similar to pure microgels made of poly(N-isopropyl acrylamide)[35,36].

**Behavior of particles at the air/water interface.** To investigate the drying and assembly behavior of the particles at the individual particle level we perform video microscopy experiments. Figure 3 compares dispersions of spherical PS reference particles (Fig. 3, top), spherical particles with a physisorbed PVA layer (Fig. 3, middle) and the ellipsoidal particles also with a physisorbed PVA layer (Fig. 3, bottom) at the start and towards the end of the drying process (Fig. 3a, b, respectively). We use image analysis to quantify the nearest neighbor distance distributions at the air/water interface (Fig. 3c). It can be seen that the spherical reference particles accumulate at the drop edge and form a close-packed structure with the nearest neighbor distance corresponding to the particle diameter (Supplementary Movie 6, Fig. 3c, top). A similar drying behavior results for elliptical particles when the physisorbed PVA shell is completely removed by washing with water/IPA (Supplementary Movie 7). Both PVA-coated particle systems preferentially adsorb to the air/water interface, forming either hexagonal arrangements (spheres, Supplementary Movie 8) or chain networks (elliptical particles, Supplementary Movie 9).

Notably, the measured nearest neighbor distances of the interfacially adsorbed particles are significantly larger (~1.6 μm) than the particle dimensions ($d = 1.1$ μm) (Fig. 3c, middle, bottom). Since the polymeric shell only increases the particle diameter in bulk by ~50 nm (Supplementary Figs. 4, 7), the large distance between particles adsorbed at the liquid interface suggests that when adsorbed at the air/water interface, the polymer chains significantly extend and spread out to maximize the interfacial area covered with polymer, which has a lower surface tension compared to plain water. This spreading leads to the formation of a very thin polymer layer surrounding the particles, which is typically termed a corona. Notably, the formation of such coronae of interfacially adsorbed particles is a well-known characteristic of core-shell particles consisting of an inorganic core and a grafted hydrogel shell adsorbed to liquid interfaces[37–39]. The interfacial polymer corona seemingly acts as a spacer layer and prevents the particles from coming into direct

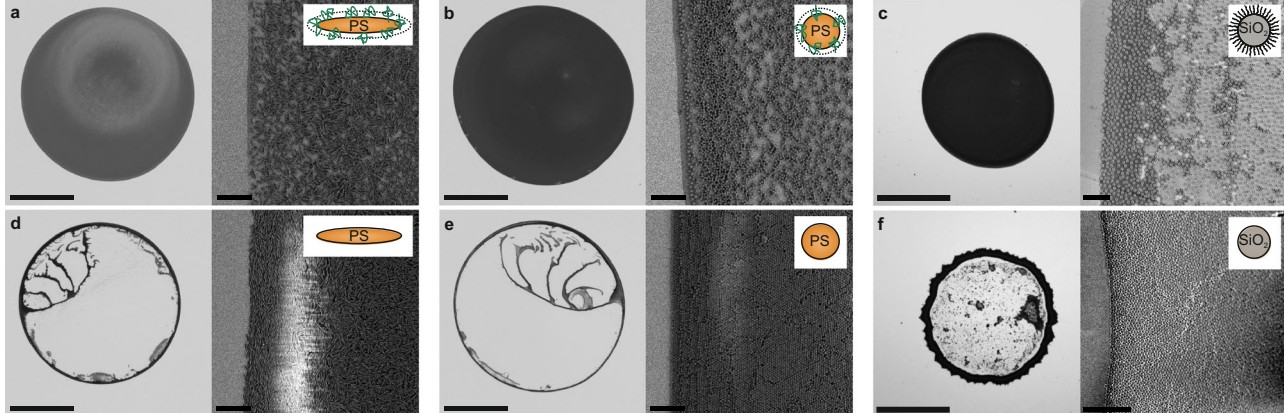

**Fig. 2 Drying behavior of pure and polymer-modified particle dispersions.** Drying patterns of aqueous dispersions containing ellipsoidal (**a**, **d**) and spherical PS particles (**b**, **e**), prepared by embedding in a PVA matrix and subsequent purification by centrifugation. Left: Optical microscopy images. Scale bar: 1 mm. Right: SEM images of the droplet edge. Scale bar: 5 μm. **a**, **b** When washed with water, both ellipsoids and spheres show a uniform drying pattern. **d**, **e** When washed with an IPA/water mixture, both dispersions show the typical coffee ring. **c**, **f** Drying behavior of a pure silica particle dispersion (**f**) and silica particles surface-functionalized with grafted linear poly(N-isopropylacrylamide) chains (**c**).

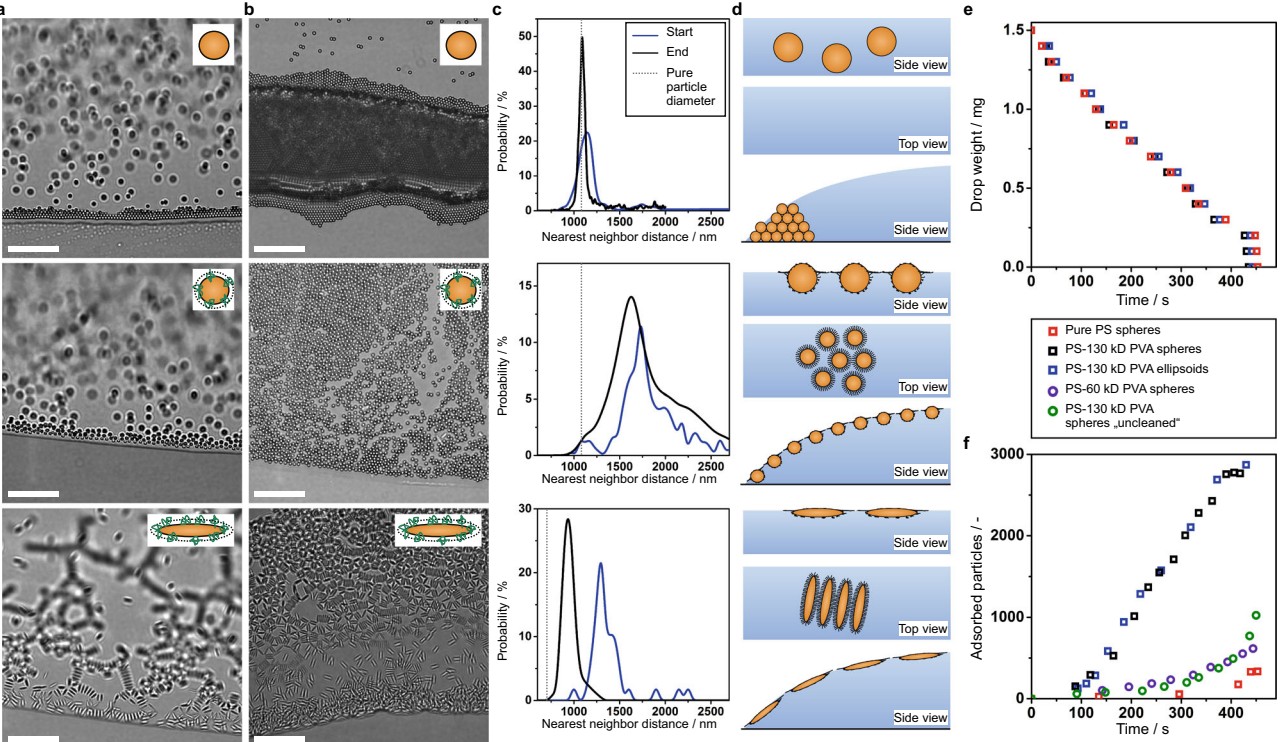

**Fig. 3 Interfacial behavior of particles during the drying of dispersions. a–d** Comparison between pure spherical PS particles (top), spherical (middle) and elliptical (bottom) PS particles with a physisorbed PVA layer (molecular weight: 130 kD). **a, b** Optical microscopy images of the contact line at the beginning (**a**) and towards the end (**b**) of the drying process. **c** Interparticle distance distribution at the start (blue) and towards the end of the drying process (black). The dotted line represents the particle diameter (sphere) and width of the ellipsoidal particles respectively without PVA. **d** Schematic illustration of the interfacial morphology of the three particl typess as side view, top view and their assembly at the contact line during the drying of the dispersion droplet. **e** Drop weight as a function of time during the evaporation process. **f** Number of adsorbed particles per area (12,500 µm²) as a function of time for the three particle systems as well as for spherical particles with a shell consisting of 60 kD PVA and a spherical particle dispersion as without the removal of excess PVA. Scale bar: 20 µm.

contact. This core-corona morphology with the polymeric chains extending along the air/water interface is schematically displayed in Fig. 3d as top view and side view.

The presence of PVA does not affect the drying rate for any particle system (Fig. 3e), indicating that the bulk fluid properties remained unaffected. However, the interfacial behavior of the particles is strongly affected by the physisorbed polymer shell (Fig. 3f). The presence of PVA at the particle surface enhances the tendency of the particles to adsorb to the liquid interface: the number of both elliptical and spherical PVA-modified particles found at the air/water interface in an optical microscope linearly increases with time, while only negligible amounts of uncoated particles are adsorbed to the interface. The molecular weight of the polymer also influences this adsorption process, with larger polymer chains being more effective at bringing the particles to the air/water interface (Fig. 3f). Also note that the presence of free polymer decreases the interfacial adsorption efficiency and only few adsorbed particles are found in an uncleaned dispersion (compare black and green data points in Fig. 3f). In agreement with the reduced surface tension of such uncleaned dispersions (Fig. 1b), this behavior indicates that free polymer occupies the liquid interface and prevents particles from adsorbing. Additionally, when ellipsoidal particles breach the air/water interface as individual particles, strong attractive quadrupolar capillary forces rapidly lead to the formation of chain networks with preferential side-to-side orientation, in agreement with previous studies[40–42] (Supplementary Movie 10, Supplementary Fig. 8). However, even in these particle chains, statistical image analysis reveals that the particles are not in direct contact, but are visibly separated,

presumably by the polymer corona, as discussed above (Fig. 3c, bottom). The occurrence of zig-zag chain patterns in SEM images after drying (Supplementary Fig. 9) further supports that picture and indicates that the interfacial arrangement was not in direct particle–particle contact. The ellipsoidal particles do not undergo any substantial Brownian motion. Spherical particles with an adsorbed PVA shell, on the other hand, undergo Brownian motion (Supplementary Movie 8, 11, Supplementary Fig. 10) and pack in a non-close packed arrangement at the air/water interface (Supplementary Fig. 11). Using particle tracking, we determine a diffusion coefficient $D = 4.46 \times 10^{-9}$ cm²s⁻¹, which agrees with theoretical calculations (Methods). Note that the Brownian motion was measured at the apex of the droplet in the absence of the Marangoni backflow, which we will discuss in the next paragraph.

**Hydrodynamic flow within drying dispersion droplets.** Next, we visualize and quantify the hydrodynamic flow within the drying dispersion droplets (Fig. 4, Supplementary Movie 12–15, Supplementary Figs. 12–15, Supplementary Discussion 2) by the addition of fluorescent tracer particles (pure and PVA-coated, respectively). For pure (Fig. 4a–d) and polymer-modified dispersions (Fig. 4e–h), we observe a radial outward bulk capillary flow and a thermal Marangoni backflow along the air/water interface, in agreement with previous publications[43–45]. While the water flow within both drying droplets appears to be similar (Supplementary Movie 12–15), the pure particles remain primarily dispersed in bulk and are therefore dominantly affected by the capillary flow (Fig. 4d). This behavior is evidenced by the

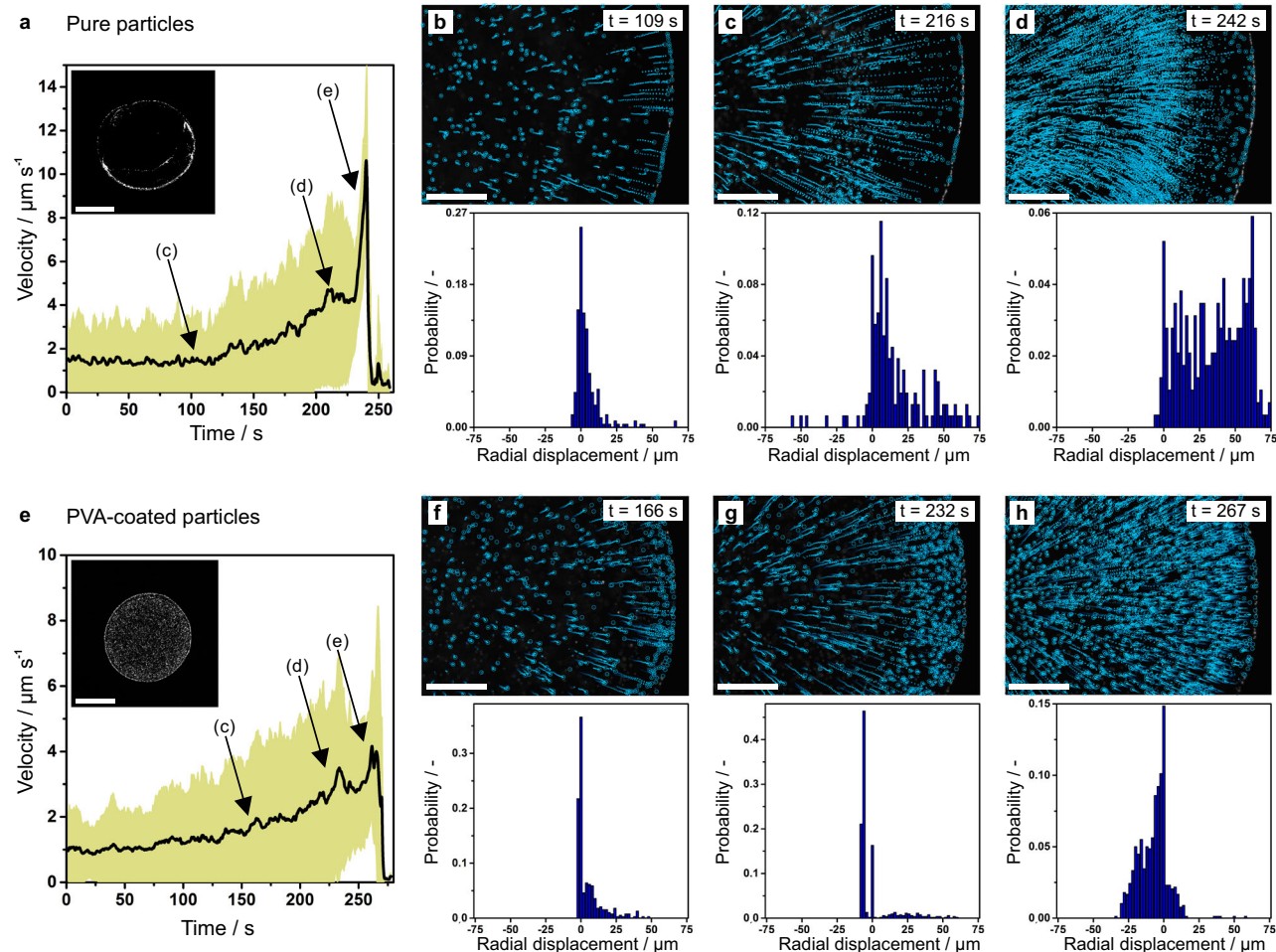

**Fig. 4 Hydrodynamic flow behavior of drying dispersions.** Hydrodynamic flow of drying of dispersions containing pure (**a–d**) and PVA-modified fluorescent PS particles (**e–h**) measured by fluorescence microscopy with the focal plane set 4 μm above the substrate. **a**, **e** Average particle velocity measured over the duration of drying, with the standard deviation shown in light green. Inset: Fluorescent microscopy image of the dried deposit. Scale bar: 1 mm. **b–d**, **f–h** Particle flow revealed by fluorescent tracer particles for pure (**b–d**) and PVA-modified particles (**f–h**). Top: Snapshots of the particle trajectory. The open circle corresponds to the position of the tracked particle and the dotted line shows the trajectory of the particles over the last 25 s. Scale bar: 200 μm. Bottom: Normalized histogram of the radial particle displacement over 5 s. Negative values correspond to an inward flow and positive values to an outward flow.

large fraction of particles that is moving outwards towards the drying edge, shown by increasingly large positive radial displacements in the statistical analysis of the tracer particles. PVA-modified particles adsorb to the air/water interface over time (Fig. 3f), do not readily agglomerate upon contact with the drying edge (Fig. 1e bottom, Supplementary Movie 8), and are therefore much more affected by the thermal Marangoni backflow, which drags the particles towards the apex of the droplet (Fig. 4h, Supplementary Movie 14, 15). This reversed movement of the polymer-modified tracer particles compared to the unfunctionalized case is obvious from the statistical analysis, which shows that the majority of particles experience a negative radial displacement. The backflow diminishes towards the apex and Brownian motion of the spherical particles is observed (Supplementary Movie 11, Supplementary Fig. 10).

**Physical interpretation of the altered drying behavior.** From the combined analysis, we deduce that the presence of the surface-active PVA chains at the particle surface influences the drying behavior in two ways. First, the polymer chains act as additional steric stabilizers that prevent effective aggregation at the droplet edge in the early stages of drying. The dominant steric repulsion

is reflected by a decrease in magnitude of the zeta potential from $(-63 \pm 5)$ mV for pure particles to $(-6.7 \pm 4)$ mV for particles with a PVA shell without negative impact on the colloidal stability. Second, the surface-active PVA chains facilitate the particle adsorption to the air/water interface, which seemingly attach irreversibly to the interface as the number of particles adsorbed to the interface continuously increases with time (Fig. 3f). The difference in the population of particles at the interface subsequently changes the movement of the particles during drying. The interfacially adsorbed particles are dragged towards the apex of the droplet by thermal Marangoni backflow and form a homogeneous interfacial particle monolayer. Once the water has evaporated, the interfacial arrangement is uniformly deposited onto the substrate. A close investigation of the dried deposit by electron microscopy reveals that the non-close packed interfacial arrangement is not preserved in the dried deposit as immersion capillary forces[46] acting on the drying particles force the particles locally into direct contact in the final stage of drying (Supplementary Figs. 9, 11). Statistical image analysis of SEM images taken through the entire dried deposit further shows a characteristic difference between spherical and elliptical particles: while both deposits appear macroscopically homogeneous, the

area fraction covered by elliptical particles slightly decreases towards the droplet interior, while it remains completely homogeneous for spherical particles (Supplementary Fig. 16, Supplementary Discussion 3). We attribute these differences to the enhanced particle interaction of the elliptical particles, which aggregate into chain-like structures and are therefore more prone to jam at the interface, while the spherical particles retain their dynamics, are efficiently dragged towards the apex and can locally equilibrate by interfacial diffusion.

**Parameters influencing the drying behavior.** The key to a successful mitigation of the coffee ring effect is the presence of surface-active polymer chains exclusively at the particle surface. Free polymer in the dispersion reduces the homogeneity of the drying pattern, as evidenced by the drying patterns of PVA/particle mixtures with successive cleaning steps (Fig. 1d, Supplementary Fig. 1) or the addition of additional PVA to the cleaned dispersion (Supplementary Fig. 17). This behavior can be rationalized by the presence of free polymer at the interface, which hinders particle adsorption to the interface, as discussed above (Fig. 3f). Additionally, the presence of surfactants is known to reduce the thermal Marangoni backflow[20,44]. This balance between free and adsorbed polymers also rationalizes previous findings on the effect of hydrosoluble polymer or ligand addition on the drying behavior of dispersions[15,30]. However, it is difficult to provide the exact amount of PVA needed to just cover the particle surface without leaving excess polymer (Supplementary Fig. 18). Similarly, adding small amounts of PVA, as required to only coat the surface, can destabilize the dispersion by bridging effects. A practical solution therefore is the addition of excess polymer followed by removal of unbound species by centrifugation. This protocol provides dispersions with most homogeneous drying behavior.

The resulting structure of the deposit depends on the particle concentration, or more precisely, the ratio between particle cross-section and the available interfacial area of the droplet (Supplementary Discussion 4, Supplementary Fig. 19). If the particle concentration is too high, not all particles are able to adsorb to the air/water interface and the excess particles experience capillary flow and thus accumulate at the droplet edge, evidenced by minor fractions at positive displacement in Fig. 4h. An insufficient number of particles leads to an incomplete area coverage with no particles at the periphery and a uniform deposition towards the center, which underlines the dominating effect of the thermal Marangoni backflow in the PVA-modified particle dispersions (Fig. 4h, Supplementary Figs. 20, 21). With appropriately adjusted concentrations, a homogenous drying pattern can be achieved independent of the droplet volume or substrate material (Supplementary Figs. 22–26). Note that these calculations need to take into account the polymer corona, which increases the interfacial particle dimensions.

We observe similar deposit patterns for different drying temperatures/drying rates, hinting at a qualitatively similar flow within the drying droplet. The quality of the deposit decreases for higher drying rates as the interfacial particles lack sufficient time to locally equilibrate positions by diffusion (Supplementary Fig. 27). Successful physisorption of PVA can be achieved for colloidal dispersions with different surface functionalities (Supplementary Fig. 28) and sizes in the colloidal regime. When the particle size approaches the polymer dimensions, agglomeration via bridging increasingly compromises the quality of the deposit (Supplementary Fig. 29).

Moreover, the presented method can be generalized to other types of polymers (Supplementary Fig. 30). A successful particle modification requires surface-activity of the polymer and

attractive polymer–particle interactions to produce a persistent polymer layer even after centrifugation and redispersion. In our case, isothermal titration calorimetry indicates attractive, predominantly hydrophobic interactions for PVA/PS particles and hydrogen bonding between PVA/hematite particles (Supplementary Fig. 31), in line with previous studies on comparable systems[47–49]. Similarly, the molecular weight of the surface-active polymers affects the drying patterns. Homogeneous drying is observed for longer polymer chains, in our case exceeding 55 kDa. In contrast, smaller polymer chains (in our case 10 kDa) are seemingly insufficient to effectively increase interfacial adsorption, as evidenced from the lower numbers of particles observed at the interface in Fig. 3f, and hence produce coffee-ring like drying patterns (Supplementary Fig. 32).

Importantly, the adsorption of surface-active polymers can fail if the particles are already sterically stabilized. If the molecular weight of the stabilizing polymer is short, such particle systems still form a coffee ring, as shown in Supplementary Fig. 33. If such sterically stabilized particles are stretched into ellipsoids, the newly formed particle surface may enable the adsorption of long surface-active polymers, which causes homogeneous drying (Supplementary Fig. 33). This additional adsorption process may also occur during the thermomechanical stretching process during the fabrication of elliptical particles and may thus provide an alternative explanation for previously observed shape-dependent drying behavior[31] (Supplementary Discussion 5).

**Versatility of the approach.** Finally, we demonstrate that the presented method is versatile with respect to the chemical nature of the particles and the dispersing liquid phase. Homogeneous drying can be induced in a range of commercial particle dispersions, such as goethite, hematite or titanium dioxide with less defined particle size and shape (Fig. 5a). These particles are not only examples of mass-produced pigments, but are also used in high-tech applications for example as functional electrodes in dye sensitized solar cells[50] or as photocatalysts in solar water splitting[51]. While the pure particle dispersions show the expected coffee ring after drying in all cases (Fig. 5b–d), a uniform drying pattern can be observed for all particle systems after surface modification with physisorbed PVA and successful removal of free PVA (Fig. 5e–g, Supplementary Figs. 34–36). The presented method can also be used to modify the drying behavior of non-aqueous systems, as long as the polymer additive decreases the interfacial tension of the medium and thus provides a driving force to enhance interfacial attraction of the modified particles (Fig. 5, Supplementary Fig. 37). While pure particles dispersed in polar liquids dry into a dot-like or coffee ring pattern (Fig. 5h–k), PVA-modified particle dispersions form homogeneous drying patterns for several organic liquids, indicating that the polymer still induces interfacial adsorption in these solvents (Fig. 5l–n). With sufficiently low surface tension, the method fails to produce homogeneous drying patterns, as the energy gain of interfacial adsorption of the polymer-coated particles becomes insufficient. In our model system, ethanolic dispersions of PVA-coated hematite, having the lowest surface tension of the tested liquids, do not produce homogeneous drying patterns (Fig. 5o).

The same underlying physical principle can be used to prevent the coffee ring effect in apolar dispersions. In this case, the chemical nature of the surface-active polymer needs to be adjusted accordingly, as we demonstrate with poly(dimethylsiloxane)-grafted-poly(methylmethacrylate) particles (Supplementary Fig. 38).

A step-by-step guide for experimentalists on choosing possible particle-polymer-solvent systems is provided in the Supplementary Note 1.

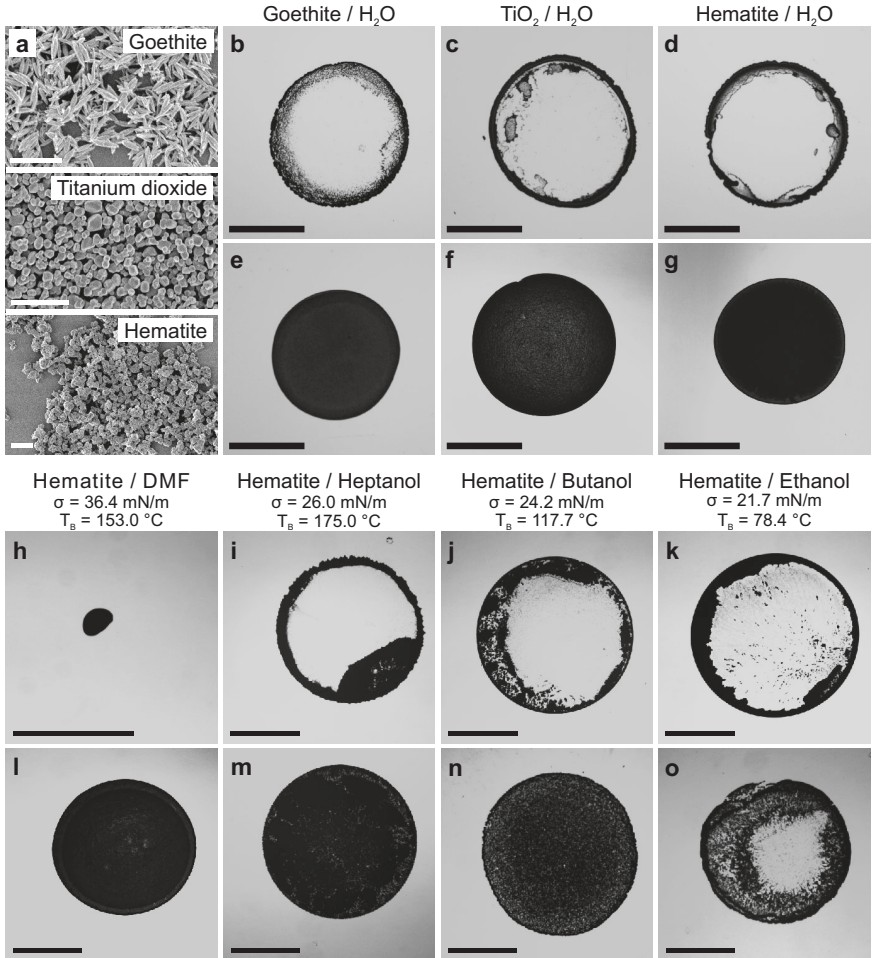

**Fig. 5 Drying behavior of pigment particle dispersions. a** SEM images of the particles. Scale bar 1 µm. **b–g** Drying behavior of aqueous dispersions of pure (**b–d**) and PVA-modified (**e–g**) goethite (**b**, **e**), titanium dioxide (**c**, **f**) and hematite (**d**, **g**) particles. **h–o** Drying behavior of pure (**h–k**) and modified (**l–o**) hematite particle dispersed in polar solvents. Scale bar: 1 mm. Aqueous dispersions are dried on a native silicon substrate (**b–g**) and dispersions in dimethyl formamide (DMF) (**h**, **l**), heptanol (**i**, **m**), butanol (**j**, **n**) and ethanol (**k**, **o**) are dried on a fluorinated substrate to ensure a well-defined sessile droplet. While the pure particles either form a coffee ring (**b–d**, **i–k**) or a dot-like deposit (caused by the high contact angle) (**h**), the PVA-coated particle dispersions dried mostly homogeneously (**e–g**, **l–n**), while for ethanolic dispersions (**o**) the coffee ring started to reappear.

## Discussion

To conclude, we demonstrate that homogeneous drying patterns of colloidal dispersions can be obtained by modifying the surface of the dispersed particles with high-molecular weight, surface-active polymers. The modification leads to enhanced steric stabilization that prevents accumulation at the drop edge and further facilitates the adsorption of the particles to the liquid/air interface. This, in turn, enhances the population of particles at the interface, which can be maximized by adjusting the total concentration of particles in the dispersion.

The important criteria to successfully prepare such a dispersion are to identify a surface-active polymer with an ability to physisorb to a desired particle system and to ensure that the dispersion remains essentially free of unbound polymer. The latter ensures that the interface is not crowded by free polymer but can be efficiently covered by the polymer-modified particles. In the course of the drying, the fluid flow of the medium remains unaffected by the modification, but the enhanced interfacial activity of the modified particles changes the net movement of the particles. Bulk particles dragged to the drying edge by capillary flows are more efficiently captured by the interface, where they experience a thermal Marangoni backflow towards the apex of the drop and therefore homogeneously distribute at the droplet surface. Homogeneous drying patterns

can thus be obtained regardless of the nature of the particles and in various liquid media, adding flexibility and versatility to control the drying of dispersions into homogeneous thin coatings.

The presented study also opens up new fundamental questions in the field of colloid and interface science Dispersions of PVA-modified particles appear to be stable even if non-adsorbed PVA is present and no signs of expected depletion attraction were observed (Supplementary Fig. 17). It would therefore be interesting to study the interaction potential between modified particles in bulk as a function of additional free polymer. In addition, it is not yet fully understood which physical-chemical properties of the adsorbed polymers are key to prevent the coffee ring effect and how the required molecular weight scales with the size of the particles to be modified. In this study, we used commercial surface-active polymers with broad molecular weight distributions. In addition, the detailed structure of the physisorbed polymer remains unknown. Future studies, using well-defined polymeric model systems, ideally with defined anchor points and grafting densities may be useful to provide more insights into the relation of the adsorbed polymer and the efficiency of interfacial capture and thus mitigation of the coffee stain effect. It is our hope that our article may trigger further research efforts in these directions.

## Methods

**Materials**. All chemicals were obtained from commercial sources and used as received if not otherwise stated. 2-[Methoxy(polyethyleneoxy)propyl]trimethoxysilane (90%, 6–9 PE units, ABCR Germany), 3-(trimethoxysilyl)propyl methacrylate (MPS, 98%, Sigma Aldrich), acrylic acid (99%, Sigma Aldrich), aminopropyltriethoxysilane (APTES, 99%, Sigma Aldrich), ammonium hydroxide solution (28–30% NH₃ basis, Sigma Aldrich), ammonium persulfate (APS, 98%, Sigma Aldrich), azobisisobutyramidine hydrochloride (97%, Wako Chemicals), azo-iso-butyronitrile (>97 %, VWR), butyl acetate (>99%, Sigma Aldrich), deuterium oxide (D₂O, 99.9%, Sigma Aldrich), dodecane (99%, Fischer Scientific), Ethanol (99.9%, Sigma Aldrich), fluorescein isothiocyanate (FTIC, 90%, Sigma Aldrich), fluorescent carboxyl-functionalized PS particles (diameter = 500 nm, Polyscience and diameter = 2 µm, Thermo Fisher), glycidyl methacrylate (97%, Sigma Aldrich), hematite particles (Bayferrox 130, Lanxess AG), hexamethyldisilazane (HMDS, >99%, Sigma Aldrich), hexane (95%, Sigma Aldrich), iron sulfate (Fe(II)SO₄, 99%, Carl Roth), isopropyl alcohol (IPA, >99.8%, Sigma Aldrich), methacrylic acid (99%, Sigma Aldrich), methanol (99.8%, Sigma Aldrich), methoxy-poly(ethylene glycol)-methacrylate (2 kDa, ICI Paints), methyl cellulose (15 cp, Sigma Aldrich), methyl methacrylate (>99%, Sigma Aldrich), Nile Red (98%, Sigma Aldrich), monomethacryloxypropyl terminated poly(dimethylsiloxane) (10 kDa, Fluorochem Ltd.), N,N-dimethylethanolamine (>99.5%, Sigma Aldrich), N,N′-methylenebis(acrylamide) (BIS, 98%, Sigma Aldrich), perfluorooctyltriethoxysilane (98%, Sigma Aldrich), Pluronics F127 (12 kDa, BASF), poly(2-ethyl-2-oxazoline) (200 kDa, Sigma Aldrich), poly(12-hydroxystearic)-glycidyl methacrylate (ICI Paints, Slough, England), poly(acrylic acid) (6 kDa, Sigma Aldrich), poly(ethylene-alt-maleic anhydride) (250 kDa, Sigma Aldrich), poly(N-Isopropylacrylamide) (PNIPAM, 80 kDa, Sigma Aldrich), polyvinyl alcohol (PVA, 130 kDa, Mowiol 18–88, Sigma Aldrich), polyvinyl alcohol (99% hydrolyzed, 200 kDa, Sigma Aldrich), polyvinyl alcohol (88% hydrolyzed, 60 kDa, Sigma Aldrich), polyvinylpyrrolidone (PVP, 360 kDa, 55 kDa, 10 kDa, Sigma Aldrich), octyl mercaptan (>98.5%, Sigma Aldrich), silica microspheres (diameter = 9.64 µm, Whitehouse Scientific), sodium carbonate (Na₂CO₃, 99%, Carl Roth), sodium dodecyl sulfate (SDS, Sigma Aldrich, 98%), sodium hydroxide (NaOH, 98%, Carl Roth), tetraethylorthosilicate (TEOS; 98%, Sigma Aldrich) and titanium dioxide (99%, VWR) were used as received.

Double-filtered and deionized water (18.2 MΩ·cm, double reverse osmosis by Purelab Flex 2, ELGA Veolia) was used throughout this study. Styrene as monomer (ReagentPlus with 4-tert-butyl catechol as a stabilizer, ≥99%, Sigma Aldrich) was washed with 10 wt-% NaOH aqueous solution to remove the inhibitor. The monomer was subsequently purified by flash column chromatography with activated Al₂O₃ (basic 90 for column chromatography, Carl Roth) using nitrogen gas. N-Isopropylacrylamide (NIPAM, 97%, Sigma Aldrich) was purified by recrystallization from hexane. Dialysis tubing (molecular weight cut off 12–14 kDa, AMS Biotechnology (Europe) Ltd) was boiled before use.

**PS particle synthesis**. The PS particles, used for the majority of the drying experiments, were synthesized by surfactant-free emulsion polymerization in a 1 L three-necked round-bottom flask with angled necks and a D-shaped stirrer at 80 rpm. First, 480 g of water was added to the flask, heated to 72 °C and purged with nitrogen for at least 30 min in order to remove oxygen. Afterwards, 20 g styrene and 0.1 g APS as initiator dissolved in 10 mL water were added consecutively with an interval of 10 min. The reaction proceeded for 24 h under nitrogen atmosphere at 72 °C. After cooling to room temperature, the dispersions were filtered using lint-free tissues (Kimtech). The colloidal dispersion was purified by dialysis against water over one month with a daily change of water.

Larger PS microspheres, used for the video microscopy observations, were synthesized by a surfactant-free emulsion polymerization. In a 500 mL triple-neck round-bottom flask, 250 mL water was heated to 80 °C and degassed by bubbling with nitrogen gas for 30 min. 80 g styrene and 0.4 g of the comonomer acrylic acid, dissolved in 5 mL water, were added under constant stirring. After 5 min, 0.1 g APS, dissolved in 5 mL water, was added. The reaction was carried out for one day at 80 °C. After cooling to room temperature, the dispersion was filtered and purified by centrifugation, redispersion and dialysis against water for two months.

The arithmetic mean size of the PS particles was characterized by measuring 100 particles in SEM images (GeminiSEM 500, Carl Zeiss AG, Germany). Small PS particles: ($d = 324 \pm 6$) nm; Large PS particles: ($d = 1100 \pm 11$) nm.

Small PS particles with acrylic acid as comonomer with a mean diameter of ($270 \pm 8$) nm were synthesized by a surfactant-free emulsion polymerization. In a 500 mL triple-neck round-bottom flask, 250 mL water was heated to 72 °C and degassed by bubbling with nitrogen gas for 30 min and stirring with D-shaped stirrer at 150 rpm. 10 g styrene and 0.1 g of acrylic acid as comonomer, dissolved in 5 mL water, were added under constant stirring. After 5 min, 0.1 g APS, dissolved in 5 mL water, was added. The reaction was carried out for one day at 80 °C. After cooling to room temperature, the dispersion was filtered and cleaned by centrifugation and redispersion in water/ethanol mixtures up to distilled water.

Amidine-functionalized PS particles with a mean diameter of ($335 \pm 105$) nm were synthesized according to Goodwin et al.[52]. The dye 7-nitrobenzo-2-oxa-1,3-diazole-methyl methacrylate was prepared as described by Jardin et al.[53] and 0.3 g of the substance was dissolved in 1 mL of methanol. 80 mL of styrene was added to 700 mL of water and the dye solution was added to this. The mixture was stirred at 350 rpm and

its temperature raised to 70 °C. After sufficient time had been allowed for the system to reach temperature equilibrium, 6 g of the initiator azobisisobutyramidine hydrochloride dissolved in 20 mL of water was added to the system. The mixture was allowed to react for 24 h and then the resultant latex was filtered through glass wool and cleaned by dialysis against distilled water and was stored in plastic containers before use.

The poly(ethylene glycol)-stabilized PS particles were prepared via dispersion polymerization as described by Harper et al.[54]. In short, 1.5 g of methoxy-poly(ethylene glycol)-methacrylate was polymerized with 30 g styrene in a water/ethanol mixture (90/315 w/w) using 1.5 g azo-isobutyronitrile as the initiator. The mixture was heated to 70 °C and refluxed for 6 h under a nitrogen atmosphere. The resultant latex particles had a mean diameter of ($647 \pm 40$) nm and were filtered through glass wool and further cleaned by repeated centrifugation and resuspension in a series of water/ethanol mixtures up to distilled water.

**Synthesis of ellipsoidal particles by mechanical stretching**. The ellipsoidal PS particles were produced according to an established protocol by Ho et al.[32]. 9 g PVA (87% hydrolyzed, 130 kDa) was dissolved in 100 mL water at 80 °C under continuous stirring. After cooling to room temperature, 4 mL of the PS particles dispersion (3 wt-%, either small or large PS particles synthesized as detailed above) was added under stirring. The dispersion was dried on a 20 × 20 cm² mirror. The resulting foil was cut into strips of 1.5 × 6 cm² and fixed at both ends. The foil was heated in an oil bath at 140 °C for 1 min and manually stretched. The film was cooled to room temperature and cleaned with a tissue soaked in IPA to remove remaining oil. The middle part of the stretched film was cut out (approx. 5 cm) and dissolved in water. The dissolved PVA was removed by three times centrifugation and redispersion in water. The ellipsoidal particles were further purified either by an additional seven times centrifugation and redispersion in water according to Yunker et al.[31] or by five times centrifugation and redispersion in IPA/water (3:7) and five times centrifugation and redispersion in water according to Ho et al.[32]. The resulting ellipsoidal particles were characterized by SEM. Mean: Small ellipsoids: ($1050 \pm 59$) nm x ($194 \pm 8$) nm, aspect ratio (AR) = $5.4 \pm 0.4$; Large ellipsoids: ($3.7 \pm 0.2$) µm x ($0.72 \pm 0.05$) µm, AR = $5.6 \pm 0.7$.

**Silica particle synthesis**. The silica particles with an mean diameter of ($294 \pm 9$) nm were synthesized via the Stöber process[55]. In a 250 mL round bottom flask, 137,5 g ethanol, 60 g water and 11.72 g NH₃(aq) were mixed, stirred and heated to 47 °C. After an equilibration time of 1 hour, 20 g TEOS was added. After 24 h, half of the particle dispersion was removed while the other half was functionalized with 50 µL MPS for an additional 24 hour.

Silica particles with a mean diameter of ($62 \pm 9$) nm were prepared by first creating a fluorescent dye by mixing 0.0357 g FITC with 0.1827 g APTES in 1.3 mL of ethanol. Three of these dye solutions were mixed with 1500 mL of ethanol, 85.7 mL of a 35% ammonia solution in water and 60 mL of tetraethylorthosilicate at room temperature. The mixture was allowed to react for 24 h. All silica particles were cleaned by repeated centrifugation and resuspension in a series of water/ethanol mixtures up to distilled water.

**Silica particles grafted with linear PNIPAM**. Linear poly(N-isopropylacrylamide) (PNIPAM) was grafted from the functionalized silica particles in a surfactant-free precipitation polymerization. In a 500 mL three-neck round bottom flask, 0.141 g NIPAM was dissolved in 47 mL water. The solution was heated to 80 °C, purged with nitrogen and equilibrated for 30 min. Meanwhile, 1.3 g aqueous silica particle dispersion (6.6 wt-%) was added. After equilibration, a balloon filled with nitrogen was used to maintain the nitrogen atmosphere and the gas inlet was removed. Subsequently, 0.011 g APS was rapidly added to initiate the reaction. After 4 hours of reaction, the modified SiO₂-PNIPAM particles were purified by centrifugation and redispersion ten times in ethanol and ten times in water.

**Goethite particle synthesis**. The ellipsoidal goethite particles were synthesized according to a customized synthesis route. In a 500 mL three-necked round bottom flask, 250 mL water was degassed with 300 mL/min nitrogen for 20 min and heated to 38 °C under continuous stirring. 15.96 g solid iron sulfate was dissolved in water after the degassing step. Then, in 50 mL degassed aqueous NaOH solution (0.21 M NaOH), 7.42 g Na₂CO₃ was dissolved and added to the iron sulfate solution under continuous stirring, inducing the precipitation of a white solid of hexagonal-shaped Fe(OH)₂ nanoplatelets[56]. The Fe(OH)₂ suspension was oxidized by pressurized air with a flowrate of 100 mL/min to form goethite. After 3 h, the reaction was stopped and the goethite particles were cleaned by three times centrifugation and redispersion in water.

**PNIPAM microgel synthesis**. The PNIPAM microgels were synthesized by precipitation polymerization following a literature protocol[57]. In a 500 mL round-bottom flask, 3.4 g NIPAM, 0.23 g BIS and 0.06 g SDS were dissolved in 195 mL water. The mixture was heated to 70 °C and degassed by bubbling nitrogen for 30 min. APS, dissolved in 5 mL water, was added to initiate the polymerization. After 4 h, the reaction was cooled to room temperature and the dispersion was purified by three times centrifugation and redispersion in ethanol and three times centrifugation and redispersion in water followed by dialysis against water for 2 weeks changing the water each day. The hydrodynamic diameter was measured by dynamic light scattering (Malvern Instruments Ltd., Zetasizer Nano ZS) to be 145 nm at 20 °C.

**Poly(methyl methacrylate) (PMMA) particle synthesis**. PMMA particles stabilized by poly(12-hydroxystearic acid)-graft-poly(methyl methacrylate) with a mean diameter of (531 ± 42) nm were prepared as described by Antl et al.[58]. Briefly, the polymeric stabilizer, poly(12-hydroxystearic)-graft-poly(methyl methacrylate), was prepared using a poly(12-hydroxystearic)-glycidyl methacrylate adduct which was reacted with methyl methacrylate and glycidyl methacrylate monomers to create a comb polymer. This was kept stored as a solution in dodecane until required. In a three-necked round-bottomed flask containing a condenser, nitrogen inlet and stirrer, 38 g of methyl methacrylate, 1.3 g of methacrylic acid, 43.8 g of hexane, 13.8 g of dodecane, 18 g of butyl acetate, 0.3 g of octyl mercaptan and 10.5 g of a 30% solution of the stabilizer in dodecane were added. Under a nitrogen atmosphere, this mixture was heated to 80 °C and stirred at 350 rpm and then an initiator solution consisting of 0.6 g of azo-isobutyronitrile dissolved in 23.7 g of methyl methacrylate was added and the polymerization reaction was allowed to proceed for 6 hours. After cooling overnight, the mixture was slowly heated up to 120 °C to remove hexane and replace it with an equivalent mass of dodecane. An amount of N,N-dimethylethanolamine equivalent to 0.2% of the total weight of the system was added to act as a catalyst in the reaction between the methacrylic acid in the particles and glycidyl methacrylate on the stabilizer which chemically locks the two together. The final latex was cleaned by repeated centrifugation and resuspension in decalin.

PMMA particles stabilized with poly(dimethylsiloxane) (PDMS) with a mean diameter of (534 ± 92) nm were prepared as follows. 0.4 g of the initiator azo-isobutyronitrile was dissolved in 10 g of methyl methacrylate monomer. 5 g of monomethacryloxypropyl terminated poly(dimethylsiloxane) (10 kDa) was dissolved in 50 g dodecane and placed in a three-necked, round bottomed flask containing a nitrogen inlet, stirrer and condenser. To this was added a further 23.6 g of dodecane, 10 g of methyl methacrylate, 0.21 g of the chain transfer agent octyl mercaptan and a solution consisting of 0.5 g of methacrylic acid containing 0.13 g of the dye Nile red. Under a nitrogen atmosphere this mixture was heated to 80 °C whilst being stirred at 350 rpm. After allowing time for temperature equilibrium to be reached, the initiator solution was added and the reaction was allowed to proceed for 6 hours. The dispersion was cleaned via repeated centrifugation and resuspension in decalin. This recipe was loosely based on Kogan et al.[59]. but we used mono vinyl terminated PDMS as comonomer to ensure a graft surface functionalization.

**Surface-functionalization of the particles with polymers**. The detailed experimental parameters for each experiment are listed in Supplementary Table 1 and SEM images of the particles used in this study are shown in Supplementary Fig. 39. Typically, 4 mL PS particles dispersed in water (diameter = 324 nm, concentration = 3.2 wt-%) were mixed with 2.56 mL PVA (87% hydrolyzed, 130 kDa) dissolved in water (5 wt-%) and kept under constant stirring for at least 2 h. The resulting physisorbed PVA content was determined to 2.3 mg/m² measured by analytical centrifugation, for which details are described below. Thus, the PVA/PS particle ratio in the mixture corresponds to an excess of polymer by a factor of 24. The mixed suspension was diluted with water to 40 mL and centrifuged at 11,000 rpm for 20 min. The supernatant was discarded and the particles were redispersed in water. The centrifugation and redispersion was repeated three times.

We tested the surface-coating of PS particles with a variety of polymers, including PVP, PVA (99% hydrolyzed, 200 kDa), PVA (88% hydrolyzed, 60 kDa), methyl cellulose, PNIPAM microgels, PNIPAM (80 kDa), Pluronics F127 (12 kDa), poly(2-ethyl-2-oxazoline) (200 kDa), poly(ethylene-alt-maleic anhydride) (250 kDa), poly(acrylic acid) (6 kDa) and SDS. The concentration was adjusted to have a similar excess in polymer and the cleaning by centrifugation was increased to five times to ensure that the dispersion was essentially free of unbound polymer.

Pigment particles including goethite (yellow pigment), hematite (red pigment) and titanium dioxide (white pigment) were functionalized with a PVA shell following the same protocol. For the goethite particles, the pH was adjusted to 10 using NaOH during the full process, to ensure that the particles have a similar negative surface charge as the substrate.

**Dispersion and solution characterization**. Pendant drop and contact angle measurements were performed with a drop shape analyzer (DSA100, Krüss). For the contact angle measurement, 4 μL of the dispersion/solutions in question were added onto a silicon wafer and dried in air. The surface tension of the aqueous dispersion/solutions was measured via the pendant drop method by depositing 8 μL droplets in air from a syringe needle (diameter: 0.5 mm). During the drying, a video was recorded for about 50 min with a frame rate of 0.093 fps. The surface tension was determined for each frame using the software provided by the instrument. The surface tension was determined after equilibration to a constant value, which typically took around 20 min. The surface tension of non-aqueous particle dispersions was measured using the Wilhelmy method setup of a KSV Nima Langmuir trough. The measurements were performed at 25 °C.

**Characterization of the drying behavior**. To characterize the drying behavior of the particle dispersions, we typically deposited 0.85 μL or 1.5 μL of each dispersion onto a silicon wafer (LG Siltron Inc. Korea), previously cleaned by sonication in IPA, ethanol and water for 5 min each. The particle concentration was typically

0.04 wt-% for PS particles and was increased for the other particles to correct for their higher density or particle size. The drying experiments were performed at 25 °C. The drying behavior was characterized by optical microscopy (Zeiss) using 1x, 2x and 2.5x objectives. To investigate the drying dynamics at the particle level, we analyzed dispersions with 1.1 μm sized PS particles on a glass slide (Carl Roth) using a 100x objective. In some cases, the contact angle of the substrates was modified by silanes. After cleaning and plasma activation (Zepto, model 2, Diener), the substrates were immersed in a 50 mL ethanolic solution containing 15 vol-% aqueous ammonia (30 vol%) and 50 μL of the silanes mPEG for 1 day, 200 μL of FOTS for 1 day or 2 mL HDMS for 3 h. A list of experimental parameters for each dried droplet can be found in Supplementary Table 1.

The number, position, motion and interparticle distance of the spherical particles were extracted from the optical image series using a custom-written particle tracking analysis software[57,60] based on the publicly available Matlab code by Crocker and Grier[61]. The ellipsoidal particles used in this study were too small to be reliably detected by the tracking software. The number of ellipsoids adsorbed to the air/water interface was counted manually using the cell counter add-on in ImageJ. Similarly, the interparticle distance for the ellipsoidal particles in a side-to-side arrangement was measured manually using ImageJ. To measure the internal flow within the droplet, fluorescent tracer particles ($d = 2$ μm, 0.005 wt-%) were added to the dispersion and imaged by fluorescence microscopy using a 10x objective.

The diffusion coefficient of the spherical particles with a PVA shell confined at the air/water interface was measured using

$$D = <x^2>/(4\tau) \tag{1}$$

where $D$ is the diffusion coefficient, $<x>$ the mean square displacement and $\tau$ the time step. The theoretical diffusion coefficient $D_T$ was calculated using

$$D_T = k_B T/(6\pi \eta r) \tag{2}$$

where $k_B$ is the Boltzmann constant, $T = 298.15$ K the absolute temperature, $\eta = 8.90 \times 10^{-4}$ Pa·s the viscosity of water at 25 °C and $r = 0.55$ μm the radius of the colloidal particle, obtaining $D_T = 4.46 \times 10^{-9}$ cm²s⁻¹.

**Characterization of the adsorbed polymeric layer**. For the determination of the anhydrous particle densities, the sedimentation properties of the bare, PVA-functionalized and IPA-washed PS particles were studied in $H_2O/D_2O$ mixtures[62] using an analytical centrifuge (LUMiSizer, LUM GmbH, Berlin) with customized cells and sector-shaped centerpieces[63]. Prior to each experiment, the particle dispersions were diluted to an optical density of about 0.7 for a 2 mm path length of the measurement cells. The particle concentration was kept identical within each $H_2O/D_2O$ series to prevent an influence of non-ideal sedimentation. Sedimentation velocity experiments were performed at 2500 rpm and 15 °C, which provided the sedimentation profiles as a function of time and space. Supplementary Fig. 5 depicts the profiles for the three different colloidal dispersions measured in three different $H_2O/D_2O$ mixtures.

The sedimentation coefficient distributions shown in Supplementary Fig. 6 were derived from the sedimentation data by least-squares direct boundary modelling and were further regularized using the Tikhonov-Phillips algorithm to stabilize the solutions against noise[64]. As significant agglomeration was observed in particular for the functionalized PS particles washed using IPA, all distributions were fitted by lognormal distributions to obtain the weight-averaged sedimentation coefficient of the slowest-sedimenting population, which relates to the primary, non-agglomerated particles. A plot of the product of sedimentation coefficient and solvent viscosity versus solution density as shown in Supplementary Fig. 6 was used to determine the anhydrous density[65,66]. Respective densities and viscosities of the liquids were taken from literature and were interpolated accordingly[67]. The anhydrous densities were determined to be 1.0513 gcm⁻³, 1.0584 gcm⁻³ and 1.0504 gcm⁻³ for the bare, PVA functionalized and IPA washed PS particles, respectively. Based on the sedimentation coefficient and the anhydrous density, the core diameter of the bare PS particles was calculated to be 322.6 nm using Stokes' equation. This is in very good agreement to the results from SEM analysis.

Next, the thickness and volume fraction of the PVA shell physisorbed onto the PS particles were determined using the experimentally derived sedimentation coefficient and anhydrous density. In the given case of a spherical particle composed of three individual constituent phases (core, ligand and solvation layer), a reasonable estimate for the composition of the particle could be obtained when combining mass conservation, volume conservation and Svedberg's equation:[68,69]

$$m_{\text{anhydrous}} = m_{\text{core}} + m_{\text{PVA}} = \rho_{\text{anhydrous}}\left(V_{\text{core}} + V_{\text{PVA}}\right) \tag{3}$$

$$V_{\text{total}} = V_{\text{core}} + V_{\text{PVA}} + V_{\text{H2O}} = m_{\text{core}}/\rho_{\text{core}} + (m_{\text{anhydrous}} - m_{\text{core}})/\rho_{\text{PVA}} + V_{\text{H2O}} \tag{4}$$

$$s = \frac{m_{\text{anhydrous}}\left(1 - \frac{\rho_s}{\rho_{\text{anhydrous}}}\right)}{f} \tag{5}$$

$m_{\text{anhydrous}}$ is the mass of the particle core and PVA shell excluding the mass of the solvation layer. The mass, volume and density of the PS core are derived from the sedimentation coefficient and anhydrous density of the bare PS particles.

The PVA volume is calculated by means of the volume fraction of the PVA in the shell $\phi_{PVA}$:

$$V_{PVA} = \phi_{PVA} V_{shell} \qquad (6)$$

Vice versa, the volume of the solvation layer is given as:

$$V_{H2O} = (1 - \phi_{PVA}) V_{shell} \qquad (7)$$

The shell volume is further determined by the particle diameter $d_{core}$ and the shell thickness $l$:

$$V_{shell} = \frac{\pi}{6}(d_{core} + 2l)^3 - V_{core} \qquad (8)$$

The mass of the PVA in the shell is calculated using the solid density of PVA, which is specified as 1.269 gcm$^{-3}$ by the supplier (Sigma Aldrich). $\rho_s$ is the density of the solvent and $f$ is the translational friction factor of the particle which is given as:

$$f = 3\pi\eta \sqrt[3]{\frac{6V_{total}}{\pi}} \qquad (9)$$

This set of equations is solved for different volume fractions of PVA and shell thicknesses. The combined relative error for the sedimentation coefficient and anhydrous density (both were measured by analytical centrifugation) is plotted in Supplementary Fig. 7. A well-defined minimum is observable which allows identification of the PVA volume fraction (4.86%) and shell thickness (30.1 nm).

With this information at hand, several properties of the PVA shell can be calculated. For example, the area occupied by one PVA chain on the PS surface is estimated to 94 nm$^2$, while its area specific mass amounts to 2.3 mgm$^{-2}$. The latter is well in line with experimental investigations on the adsorption isotherms of different PVA species on PS particles[33]. Chibowski found area specific masses between 2.76 mgm$^{-2}$ and 4.23 mgm$^{-2}$ for PVA with molecular weights ranging from 36 to 130 kDa adsorbing on 900 nm PS particles. Slight deviations in the PVA coverage in the present work can be due to different surface properties (hydrophilicity, surface charge, functional groups) of the PS particles or are a result of the different core size.

**Isothermal titration calorimetry (ITC).** Heatflow measurements were performed using a TAM III thermostat (TA Instruments) equipped with a nanocalorimeter (TA Instruments). The reference cell contained 1 mL water and the sample cell contained either 1 mL water as reference, 1 mL PS dispersion (4 wt-%) or 1 mL hematite suspension (16 wt-%). For the measurement, 20 aliquots with a volume of 20 μL PVA solution (0.14 wt-%) were injected from a 500 μL Hamilton glass syringe under constant stirring at 80 rpm. The temperature of the ITC device was kept constant at 25 °C and the equilibration time was 1 h after each injection.

The normalized heats were analyzed with a fitting procedure according to an independent binding model[47–49] (Eqs. (10), (11)) to obtain the reaction enthalpy $\Delta H$, the association constant $k_a$ and the reaction stoichiometry $n$ as the fitting parameters,

$$Q = \frac{n[Part]\Delta H V_0}{2}\left[1 + \frac{[PVA]}{n[Part]} + \frac{1}{nk_a[Part]} - \sqrt{\left(1 + \frac{[PVA]}{n[Part]} + \frac{1}{nk_a[Part]}\right)^2 - \frac{4[PVA]}{n[Part]}}\right] \qquad (10)$$

$$\Delta Q_i = Q_i + \frac{dV_i}{V_0}\left[\frac{Q_i + Q_{i-1}}{2}\right] - Q_{i-1} \qquad (11)$$

where $Q$ is the total heat content, $[Part]$ the particle mass concentration, $[PVA]$ the PVA mass concentration and $V$ the active cell volume. The Gibbs free energy $\Delta G$ is calculated by Eq. (12),

$$\Delta G = RT \ln(k_a) \qquad (12)$$

where $R$ is the universal gas constant and $T$ the temperature.

## Data availability
The data generated in this study (analytical centrifuge, dynamic light scattering, pendant drop, contact angle, image analysis) have been deposited at zenodo (https://doi.org/10.5281/zenodo.6448185).

## Code availability
The matlab code used for image analysis are available at zenodo (https://doi.org/10.5281/zenodo.6448405).

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

# ARTICLE

35. Mayarani, M., Basavaraj, M. G. & Satapathy, D. K. Loosely packed monolayer coffee stains in dried drops of soft colloids. *Nanoscale* **9**, 18798–18803 (2017).

36. Horigome, K. & Suzuki, D. Drying mechanism of poly(N-isopropylacrylamide) microgel dispersions. *Langmuir* **28**, 12962–12970 (2012).

37. Rauh, A. et al. Compression of hard core-soft shell nanoparticles at liquid-liquid interfaces: influence of the shell thickness. *Soft Matter* **13**, 158–169 (2016).

38. Rey, M., Fernandez-Rodriguez, M. A., Karg, M., Isa, L. & Vogel, N. Poly-N-isopropylacrylamide nanogels and microgels at fluid interfaces. *Acc. Chem. Res.* **53**, 414–424 (2020).

39. Menath, J. et al. Defined core–shell particles as the key to complex interfacial self-assembly., 118(52). *Proc. Natl Acad. Sci.* **52**, 118 (2021).

40. Loudet, J. C., Alsayed, A. M., Zhang, J. & Yodh, A. G. Capillary interactions between anisotropic colloidal particles. *Phys. Rev. Lett.* **94**, 2–5 (2005).

41. Lim, J. H. et al. Heterogeneous capillary interactions of interface-trapped ellipsoid particles using the trap-release method. *Langmuir* **34**, 384–394 (2018).

42. Botto, L., Lewandowski, E. P., Cavallaro, M. & Stebe, K. J. Capillary interactions between anisotropic particles. *Soft Matter* **8**, 9957–9971 (2012).

43. Ristenpart, W. D., Kim, P. G., Domingues, C., Wan, J. & Stone, H. A. Influence of substrate conductivity on circulation reversal in evaporating drops. *Phys. Rev. Lett.* **99**, 1–4 (2007).

44. Marin, A., Liepelt, R., Rossi, M. & Kähler, C. J. Surfactant-driven flow transitions in evaporating droplets. *Soft Matter* **12**, 1593–1600 (2016).

45. Deegan, R. D. et al. Contact line deposits in an evaporating drop. *Phys. Rev. E - Stat. Phys., Plasmas, Fluids, Relat. Interdiscip. Top.* **62**, 756–765 (2000).

46. Vogel, N., Retsch, M., Fustin, C. A., Del Campo, A. & Jonas, U. Advances in colloidal assembly: the design of structure and hierarchy in two and three dimensions. *Chem. Rev.* **115**, 6265–6311 (2015).

47. Winzen, S. et al. Complementary analysis of the hard and soft protein corona: sample preparation critically effects corona composition. *Nanoscale* **7**, 2992–3001 (2015).

48. Prozeller, D., Morsbach, S. & Landfester, K. Isothermal titration calorimetry as a complementary method for investigating nanoparticle-protein interactions. *Nanoscale* **11**, 19265–19273 (2019).

49. Chiad, K. et al. Isothermal titration calorimetry: a powerful technique to quantify interactions in polymer hybrid systems. *Macromolecules* **42**, 7545–7552 (2009).

50. Sauvage, F. et al. Dye-sensitized solar cells employing a single film of mesoporous TiO 2 beads achieve power conversion efficiencies over 10%. *ACS Nano* **4**, 4420–4425 (2010).

51. Sivula, K., Le Formal, F. & Grätzel, M. Solar water splitting: progress using hematite (α-Fe 2O3) photoelectrodes. *ChemSusChem* **4**, 432–449 (2011).

52. Goodwin, J. W., Ottewill, R. H. & Pelton, R. Studies on the preparation and characterization of monodisperse polystyrene latices V.: The preparation of cationic latices. *Colloid Polym. Sci.* **257**, 61–69 (1979).

53. Jardine, R. S. & Bartlett, P. Synthesis of non-aqueous fluorescent hard-sphere polymer colloids. *Colloids Surf. A Physicochem. Eng. Asp.* **211**, 127–132 (2002).

54. Harper, G. R. et al. Sterie stabilization of microspheres with grafted polyethylene oxide reduces phagocytosis by rat Kupffer cells in vitro. *Biomaterials* **12**, 695–700 (1991).

55. Stöber, W., Fink, A. & Bohn, E. Controlled growth of monodisperse silica spheres in the micron size range. *J. Colloid Interface Sci.* **26**, 62–69 (1968).

56. Encina, E. R., Distaso, M., Klupp Taylor, R. N. & Peukert, W. Synthesis of goethite α-FeOOH particles by air oxidation of ferrous hydroxide Fe(OH)2 suspensions: Insight on the formation mechanism. *Cryst. Growth Des.* **15**, 194–203 (2015).

57. Rey, M., Law, A. D., Buzza, D. M. A. & Vogel, N. Anisotropic self-assembly from isotropic colloidal building blocks. *J. Am. Chem. Soc.* **139**, 17464–17473 (2017).

58. Antl, L. et al. The preparation of poly (methyl methacrylate) latices in non-aqueous media. *Colloids Surf.* **17**, 67–78 (1986).

59. Kogan, M., Dibble, C. J., Rogers, R. E. & Solomon, M. J. Viscous solvent colloidal system for direct visualization of suspension structure, dynamics and rheology. *J. Colloid Interface Sci.* **318**, 252–263 (2008).

60. Rey, M. et al. Isostructural solid–solid phase transition in monolayers of soft core–shell particles at fluid interfaces: structure and mechanics. *Soft Matter* **12**, 3545–3557 (2016).

61. Crocker, J. C. & Grier, D. G. When like charges attract: the effects of geometrical confinement on long-range colloidal interactions. *Phys. Rev. Lett.* **77**, 1897–1900 (1996).

62. Nontapot, K., Rastogi, V., Fagan, J. A. & Reipa, V. Size and density measurement of core–shell Si nanoparticles by analytical ultracentrifugation. *Nanotechnology* **24**, 155701 (2013).

63. Uttinger, M. J. et al. New prospects for particle characterization using analytical centrifugation with sector-shaped centerpieces. *Part. Part. Syst. Charact.* **37**, 2000108 (2020).

64. Walter, J., Thajudeen, T., Süß, S., Segets, D. & Peukert, W. New possibilities of accurate particle characterisation by applying direct boundary models to analytical centrifugation. *Nanoscale* **7**, 6574–6587 (2015).

65. Brown, P. H., Balbo, A., Zhao, H., Ebel, C. & Schuck, P. Density contrast sedimentation velocity for the determination of protein partial-specific volumes. *PLoS One* **6**, e26221 (2011).

66. Fagan, J. A. et al. Analyzing surfactant structures on single-wall carbon nanotubes by analytical ultracentrifugation. *ACS Nano* **7**, 3373–3387 (2013).

67. Cho, C. H., Urquidi, J., Singh, S. & Robinson, G. W. Thermal offset viscosities of liquid H2O, D2O, and T2O. *J. Phys. Chem. B* **103**, 1991–1994 (1999).

68. Benoit, D. N. et al. Measuring the grafting density of nanoparticles in solution by analytical ultracentrifugation and total organic carbon analysis. *Anal. Chem.* **84**, 9238–9245 (2012).

69. Carney, R. P. et al. Determination of nanoparticle size distribution together with density or molecular weight by 2D analytical ultracentrifugation. *Nat. Commun.* **2**, 335–338 (2011).

## Acknowledgements
The project was funded by the Deutsche Forschungsgemeinschaft (DFG, German Research Foundation) – Project-ID 416229255 – SFB 1411. N.V. also acknowledges funding by the Deutsche Forschungsgemeinschaft under grand numbers VO1824/6-2 and VO1823/8-1. W.P. funding by the Deutsche Forschungsgemeinschaft under grand number PE427/33-1. M.R. acknowledges funding from the Swiss National Science Foundation Project-ID P2SKP2_194953. C.M.P. acknowledges studentship funding from the EPSRC Centre for Doctoral Training in Soft Matter and Functional Interfaces SOFI-CDT, EP/L015536/1. M.R. acknowledges Paul Clegg as host in Edinburgh and temporary funding from EAM. We acknowledge the use of the SEM bought with the EPSRC grant EP/P030564/1. The authors thank Svenja Morsbach for discussions of the ITC data.

## Author contributions
M.R. modified the particle surface with polymers and studied the drying behavior of the dispersions. M.R., J.W., A.F., M.J.U. and W.P. characterized the adsorbed polymer layer by analytical centrifugation. S.C. and A.B.S. synthesized the PS particles, S.N. synthesized the ellipsoidal particles under supervision of M.R., M.I. synthesized the silica and grafted silica particles, M.M. and M.D. synthesized the goethite particles. A.B.S and M.R synthesized PMMA particles. S.C. and M.R. measured the surface tensions and contact angles, M.R. evaluated the drying data by optical microscopy, confocal microscopy and M.R and J.H. measured the particle diffusion at the liquid interface. C.M.P., J.H.J.T. and M.R. performed the internal flow-analysis. M.D., M.J.U., M.R. and N.V. performed the ITC analysis. M.R. and N.V. designed the experiments and wrote the manuscript with contributions of all other authors. N.V. supervised the study.

## Funding

## Competing interests
The authors declare no competing interests.
