## [Peer review file · Nature Communications]

REVIEWERS' COMMENTS

Reviewer #2 (Remarks to the Author):

The authors report very interesting observations on how adsorption of polymers alter the drying behaviour of a droplet on a surface. Moreover, these observations are supported by experiments testing the hypothesis presented and provide a physical understanding on how coffee stain is suppressed. Furthermore, this study sheds light to observations of Yunker et al published previously. I strongly recommend publication of this manuscript after minor points given below are addressed.

1) In the abstract, authors say "After spilling coffee, a tell-tale circular stain is left by the drying droplet ..." The coffee stains in real life surfaces that are a far cry from the well-cleaned glass slides used in this study. Consequently, the coffee stains on these heterogenous surfaces are rarely circular. I recommend removing the word circular as this will help readers in industrial practice.

2) The authors need to clarify what are the open questions left. I recommend addition of a short discussion before conclusion section where authors bluntly discuss the open questions and limitation of the proposed method as this will increase the impact of their work. Here is what I can think of to help the authors: What is the critical concentration, MW, chemical identity of the polymer required to suppress coffee stains? What is the most dense suspension by volume fraction this effect would work? Is there a concentration range where depletion interactions dominate or stability?

Reviewer #3 (Remarks to the Author):

I again thank the authors for their thorough responses. I believe that the manuscript benefits from the expanded results and discussion.

I found only one outstanding issue. As pointed out before, Yunker et al. also prepared spheres following the same protocol as ellipsoids, absent stretching. As written, the sentence "Surprisingly, however, a sample with spherical..." (line 76) creates a false impression that this approach was unique to the current manuscript when it is not. Perhaps this sounds pedantic, but it is important to understanding the physics uncovered here; a major part of the mystery is why two sets of particles created with the same protocol exhibited similar behaviors in some cases, but different behaviors in others. As written, the text creates the false impression that the approaches were different when in fact they were not, and must be clarified.

REVIEWERS' COMMENTS

Reviewer #2 (Remarks to the Author):

The authors report very interesting observations on how adsorption of polymers alter the drying behaviour of a droplet on a surface. Moreover, these observations are supported by experiments testing the hypothesis presented and provide a physical understanding on how coffee stain is suppressed. Furthermore, this study sheds light to observations of Yunker et al published previously. I strongly recommend publication of this manuscript after minor points given below are addressed.

1) In the abstract, authors say "After spilling coffee, a tell-tale circular stain is left by the drying droplet ..." The coffee stains in real life surfaces that are a far cry from the well-cleaned glass slides used in this study. Consequently, the coffee stains on these heterogenous surfaces are rarely circular. I recommend removing the word circular as this will help readers in industrial practice.

As suggested by Rev. 2, we removed the word "circular" from the first sentence, which now reads as:

After spilling coffee, a tell-tale stain is left by the drying droplet. This universal phenomenon, known as the "coffee ring effect", is observed independent of the dispersed material.

2) The authors need to clarify what are the open questions left. I recommend addition of a short discussion before conclusion section where authors bluntly discuss the open questions and limitation of the proposed method as this will increase the impact of their work. Here is what I can think of to help the authors: What is the critical concentration, MW, chemical identity of the polymer required to suppress coffee stains? What is the most dense suspension by volume fraction this effect would work? Is there a concentration range where depletion interactions dominate or stability?

As suggested by Rev. 2, we added an in-depth discussion at the end of the manuscript, where we address the open questions and guide fellow scientist on how to tackle them. The discussion chapter reads as:

The presented study also opens up new questions in the field of colloid and interface science. Suspensions of PVA-modified particles appear to be stable even if non-adsorbed PVA is present and no signs of expected depletion attraction were observed (Supplementary Figure 17). It would therefore be interesting to study the interaction potential between modified particles in bulk as a function of additional free polymer. In addition, it is not yet fully understood which physical-chemical properties of the adsorbed polymers are key to prevent the coffee ring effect and how the required molecular weight scales with the size of the particles. In this study, we used commercial surface-active polymers with broad molecular weight distributions. In addition, the detailed structure of the physisorbed polymer remains unknown. Future studies, using well-defined polymeric model systems, ideally with defined anchor points and grafting densities may be useful to provide detailed insights. It is our hope that our article may trigger further research efforts in these directions.

Reviewer #3 (Remarks to the Author):

I again thank the authors for their thorough responses. I believe that the manuscript benefits from the expanded results and discussion.

I found only one outstanding issue. As pointed out before, Yunker et al. also prepared spheres following

the same protocol as ellipsoids, absent stretching. As written, the sentence "Surprisingly, however, a sample with spherical..." (line 76) creates a false impression that this approach was unique to the current manuscript when it is not. Perhaps this sounds pedantic, but it is important to understanding the physics uncovered here; a major part of the mystery is why two sets of particles created with the same protocol exhibited similar behaviors in some cases, but different behaviors in others. As written, the text creates the false impression that the approaches were different when in fact they were not, and must be clarified.

As suggested by Rev. 3, we changed the beginning of the sentence to avoid a false impression that this approach was unique to the current manuscript. The sentence now reads as:

In contrast to previous findings, however, a sample with spherical particles prepared by the same protocol, without the stretching step, shows a similar, uniform drying pattern (Figure 2b, Supplementary Movie 3).